# Procedure of Numerical Modelling and Estimation of Sieve Curve Changes as a Tool to Define Riverbed's Erodibility

**Jacek Florek *** and **Maciej Wyrębek**

Department Water Engineering and Geotechnics, Agriculture University, Al. A. Mickiewicza 24/28, 30-059 Cracow, Poland
* Correspondence: jacek.florek@urk.edu.pl; Tel.: +48-12-662-4172

**Abstract:** The numerical 1-D HEC-RAS modelling tool was supported by the estimation of the sieve curve changes procedure to measure the scale of predicted discharges along a stretch of stream in southern Poland on the Olkusz Upland. The procedure was calibrated in southern Poland on the mountain streams during high-stage events, using a radiotracer application in bedload transport. Particular terrain hypsometry, created by the dissolution of limestone, forced the deep erosion of the river valley bottom; it is here that the current shape of the riverbed of the Prądnik stream is placed. While numerical modelling is widely used in hydraulics, standards have been set for the estimation of flood risk zones; these estimations suggest that the densities of the measured cross-sections are less then optimal, and that the erosive processes are more frequent. This was proved by identifying a number of erosive sections. A new procedure proposed combining the prediction of grain size distribution with hydraulic modelling. Calculations using the estimation of sieve curves, based on the processes of creation and destruction in the armouring layer, have proven to be a challenge for the existing standards of hydraulic modelling. We believe that it is easy to expand the usefulness of the 1D model by utilising its results for this procedure. For the purpose of this type of analysis, dense cross-section measurements are involved, careful modelling is required and a wide range of additional in-field data has to be gathered. For the interpretation of the results, the relation between channel-forming discharge, bankfull discharge, present and critical shear stresses, as well as the mean diameter of the grain size and other estimated sieve curve parameters, were evaluated. Channel-forming discharge is smaller than the bankfull discharge in more than one third of the segment where the erosion process is more frequent and the stability of the riverbed is compromised. Channel-forming discharge was at least twice as high in the stable sections, compared to the erosive section. The presented method will help to find unstable riverbed sections, in order to mitigate the dimension of river training techniques and protect the natural state of the river. While we are in the period of development in this region of Europe, limiting the scope of interference in rivers and streams by applying this method may create an opportunity for the concept of river training close to nature.

**Keywords:** channel-forming discharge; erosion; sieve curve estimation; 1-D hydraulic model

## 1. Introduction

When using numerical models that allow one to determine the hydraulic conditions of water movement, it is typical to operate the data that concern the whole catchment area, together with tributaries. Then, the modelling of significant lengths of rivers segments takes place where the specificity of calculations mainly depends on the range of the performed task and the character of the analysed water course. In such cases, the main task of the modelling is to determine the level of the water surface (in the 1D model), as well as flow directions (in the 2D model), velocity and the forces of water acting on obstacles in the 2D model. Next, the obtained results will serve to generate the zones of flood risk or to modify the area geometry with water movement. The analyses of such a type include, to a small

extent, the stability conditions of the riverbed. The assumed values that concern the bed [1] can stay constant in the whole range. However, the roughness values can be variable in the conditions of flooding [2–6] and the bottom can be reshaped as a result of erosion and/or accumulation [7,8]. Calculations that allow one to determine the estimation of changes in the bottom's sieve curve, in the case of flood passing, have been previously performed using the classic formulas [9]. In Polish conditions, the procedure was calibrated in the south of the country on the mountain streams, using a radiotracer application in bedload transport [4,10]. However, in the face of the current access to the mutually conditional hydraulic parameters of water movement, provided by numerical modelling, a renewed and much more detailed approach [11] to the issue, connected to the estimation of the stability of a riverbed [12] under the influence of flood, became possible.

The basic tools that serve to realize such a task are as follows:

- Geometric measurements in field (it is possible to support them with the numerical models of terrain; however, the land that is under the water, as well as the ground densely overgrown with plants, does not provide complete and reliable data);
- A hydraulic measurement (velocity and discharge);
- An analyses of spatial data (complementary information, verification of measure data, elimination of errors);
- A collection of bottom granulation samples of the watercourse bed and their analysis (sieve curves);
- Creating a numerical model for the hydraulic conditions of water movement;
- A calibrating process;
- A computational estimation of granulation changes;
- An interpretation of the results, including the indication of the segments susceptible to erosion and those with a predominant accumulation;
- A spatial relation between erosive and accumulative segments, and the morphology of the modelled part of the river.

The presented calculation methodology concerns a segment of the Prądnik stream, between the mouths of the Sąspówka and Korzkiewka.

## 2. Methodology

### 2.1. Research Area

Prądnik is a left-bank tributary of the Vistula river with a catchment area located in Małopolska, in the area of the Olkusz Upland. The length of the watercourse is 35.9 km, and the catchment area is 193 km$^2$ [13]. Low water states dominate in January and February, as well as in November and December, and the culminations are in March and April. The share of underground water supply is high (at approximately 75%), and the one-hundred-year flood in the research stream is 77 m$^3$s$^{-1}$. The surveyed segment is situated in the central part of the catchment area between the mouths of two tributaries (Figure 1.)

Geological bedrock is composed of limestones and upper Jurassic marls. An increasing indent in the bottom of the Prądnik valley starts in the area of Pieskowa Skała (in English: *Little Dog's Rock*); it reaches maximal values of 120 m at the area of the Ojców National Park and then decrease up to Giebułtów. The reasons for the indent in the valley's bottom are placed in the Jurassic rocky limestones, susceptible to cracks and crevices, that lead to creating the karstic forms [9]. Tortuosity and the significant diversity of the valley's bottom is a result of several factors: the various paces of erosive processes and the dissolution of the bottom rocks, landslides of the slopes' material, provision from the side valleys, and the creation of cone-shaped fans at their numerous mouths. This process concerns not only perennial tributaries, such as Sąspówka and Korzkiewka, but also those occurring temporarily that join Prądnik in the tested segment. The dynamics of these processes was, in the past, sufficient to create karstic ravines that resulted in a rock gorge, e.g., the Kraków Gate. All these phenomena, together with the predominant erosive activity of Prądnik itself, led to the creation of an irregular structure at the valley's bottom. Typical for Polish

conditions, altitude systems of flood terraces cause the very rare access of flood waters to them [14–16]; however, being narrow, deeply indented and located in direct contact with the watercourse, the terrace of Prądnik can be more frequently influenced by the erosive activity of water.

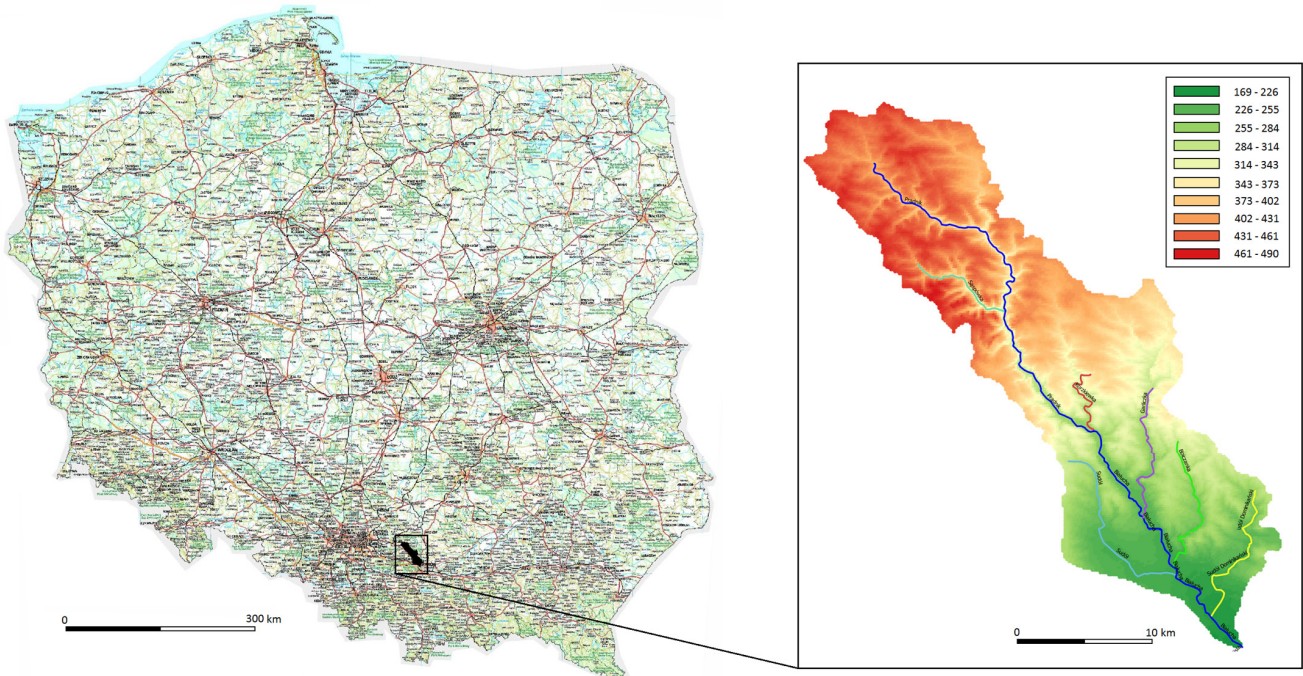

**Figure 1.** Location of Prądnik catchment area (TOPO http://mapy.geoportal.gov.pl accessed on 2 December 2022, DMT http://www.codgik.gov.pl).

### 2.2. Measurement

The surveys carried out in Prądnik consisted of measuring the routes of the stream's cross-sections in the field, using the total station on the basis of a points canvas, built using the GPS RTK (Global Positioning System, Real Time Kinematic, vertical accuracy $v_{avg} = 0.03$, $v_{max} = 0.05$ m) device. Measurements of granulation were performed by means of the sifting technique in selected points, according to the changes occurring in the bottom's cover. Prądnik is characterized by certain variabilities in morphological structure, and in order to represent it on the analysed part of the riverbed, five locations for the measurement of granulation were selected. The largest grain size $d_{max}$ in the sample never exceeded 5% of the total mass.

To provide a good projection of the conditions of the water movement in the numerical model, and for the larger valley flows, on-site verification was performed to determine the indexes of roughness for the floodplain area, based on the plant cover. In total, 61 cross-sections were measured altogether along the measurement segment of 4600 m. In the course of repeated measurement sessions, the concentration of the cross-sections was constantly extended; this was caused by the interpretation of the previous results, of measurements and calculations, demonstrating the diversity of the riverbed and the presence of segments that require the provision of more details [17–19].

The values of flow were obtained from the gauging station located in Ojców.

Measuring the conditions of Prądnik is difficult, due to the significant indent in the valley's bottom and the weak exposition in the satellite measurements; in addition, there is poor local communication in the ground system between referential stations within the GSM network, and access to fenced places or areas extensively covered with plants is hindered. Measurements in the field should preferably be performed in fine weather so that there is little sky cover, beyond the period that vegetation is excessive and when the water level in Prądnik is low.

### 2.3. Modelling of Hydraulic Conditions

The calculations for the hydraulic parameters of water movement were performed by means of the one-dimensional hydraulic model HEC-RAS; this serves to calculate steady and unsteady flow conditions in rivers and water reservoirs on the basis of the equation for energy balance, or the system of two partial differential equations, provided by Saint-Venant (www.hec.usace.army.mil, accessed on 2 December 2022) [20].

The water surface profiles were computed from one cross section to the next by solving the energy equation with an iterative procedure called the standard step method. The energy equation is written as follows:

$$Z_2 + Y_2 + \frac{a_2 v_2^2}{2g} = Z_1 + Y_1 + \frac{a_1 v_1^2}{2g} + h_e$$

where

$Z_1$, $Z_2$—elevation of the main channel inverts
$Y_1$, $Y_2$—depth of water at cross sections [m],
$a_1$, $a_2$—velocity weighting coefficients,
$g$—gravitational acceleration [$ms^{-2}$].
$h_e$—energy head loss [m].
The energy head loss ($h_e$) between two cross-sections is comprised of friction losses and contraction or expansion losses. The equation for the energy head loss is as follows:

$$h_e = L \cdot \overline{S}_f + C \left[ \frac{a_2 v_2^2}{2g} - \frac{a_1 v_1^2}{2g} \right]$$

where L—discharge weighted reach length [m],
$\overline{S}_f$—representative friction slope between two sections,
C—expansion or contraction loss coefficient.
The preparation of the model's structure needs supplementing and the selection of proper flows is required; in case of diversified cross-sections, these had to reflect the full range of the potential riverbed capacities of the segment [21]. The model was calibrated using the measurement of water levels in cross-sections and data from the gauge station. The gauge station in Ojców ($Q_{1\%}$ = 31 $m^3 s^{-1}$, www.kzgw.gov.pl, accessed on 16 October 2022) is located 1km upstream from the segment and there are no tributaries on this section. The calculations were performed for 300 various flow values in the range of 0.06–470 $m^3 s^{-1}$.

### 2.4. Estimation of the Sieve Curve Changes

Based on the results obtained by model calculations, using the calculation procedure that determines the initial conditions of the beginning of the mass bedload transport [22], on the bottom of Prądnik, became possible [23]. The results of the model calculations for the given flow sizes allowed the mutual linkage of the conditioned parameters of individual cross-sections for the given flows [24]; this enables the full use of the procedure available in the programme ARMOUR [4,10]. If the massive bedload transport appears in the riverbed before discharge approaches the flood plain, the cross-section should be considered to be erosive. The frequency of this discharge plays an important role. This is because, in the large cross-sections (also those with small slopes), a very large flow, i.e., with a low frequency of appearance, is needed to give rise to strong erosion. However, when the debris movement takes place before the flood of waters to the valley's area (already in the waterbed), then this phenomenon refers to significantly lower and more frequent flows; therefore, the process of erosion will be common, even in this part of the riverbed.

The described calculation procedure requires proper preparation of the series of calculation data, including the following:

- ordinates of the water surface in the range and amount that provides the required accuracy, including all possible discharges for the set of cross-sections in the single calculation procedure,
- energy line slopes equivalent to the ordinates of the water surface,
- roughness values,
- parameters of cross-sections,
- sieve curves.

The effect of the performed calculations are the values of changing hydraulic parameters, including the following:

- estimated changes in the bottom's sieve curve,
- values of real and critical shear stresses,
- standard deviation of the sieve curve.

Using calculations, including the values of dimensionless stresses ($f_i$) of individual fractions ($d_i$), critical shear stresses at the beginning of movement can be calculated [9,25].

$$\tau_{gr} = f_i \, g \, \Delta\rho_s \, d_i$$

where:

$f_i$—dimensionless stresses for the i-th fraction [-],
g—gravitational acceleration [$ms^{-2}$],
$\Delta\rho_s$—difference between the densities of the debris and the water [$kgm^{-3}$],
$d_i$—diameter of the i-th fraction [m].

The interpretation of the results is based on determining the changes in the sieve curve. The size of water forces acting on the bottom increases together with the growing flood, causing the gradual movement of debris. The effect of this phenomenon is the change in the content of individual fractions in the sieve curve, so called the sorting of the material. The growing movement of debris and the course of the granulation curve moves down on the graph [2–4,22]. This strongly erosive process continues until a significant level in an armouring layer of the riverbed is reached; it eventually ends when still-growing forces in the bottom lead to the erosive destruction of the armouring layer, which causes the mass movement of debris.

The proper interpretation of the process can be performed with the simultaneous analysis of four values: real and critical shear stresses, the standard deviation of the sieve curve and the representative value of the granulation diameter $d_m$.

## 3. Results

### 3.1. RiverBed Geometry

Prądnik, between the mouths of Sąspówka and Korzkiewka, flows and creates meandering segments. Frequently transferring their location in the valley, the cross-sections present variable conditions of slope and shape that indicate the variability of the erosive processes occurring there.

Figure 2 presents the fragments of the cross-sections horizontal position that reflects the changes in directions and translations in the valley. Apart from determining the level of the channel-forming discharge, the longitudinal format of the riverbed also provides information about the dynamics of the phenomena at the bottom of Prądnik (Figure 3).

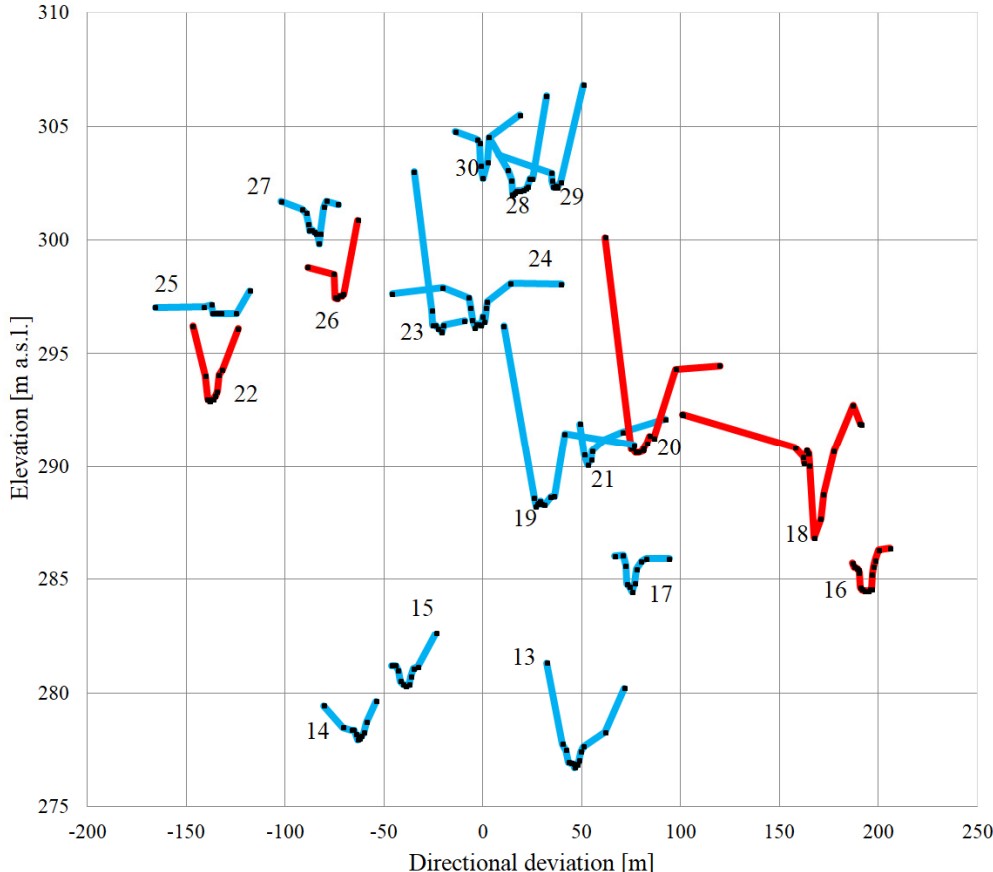

**Figure 2.** Low water parts of cross-sections (red-erosive, blue-stable).

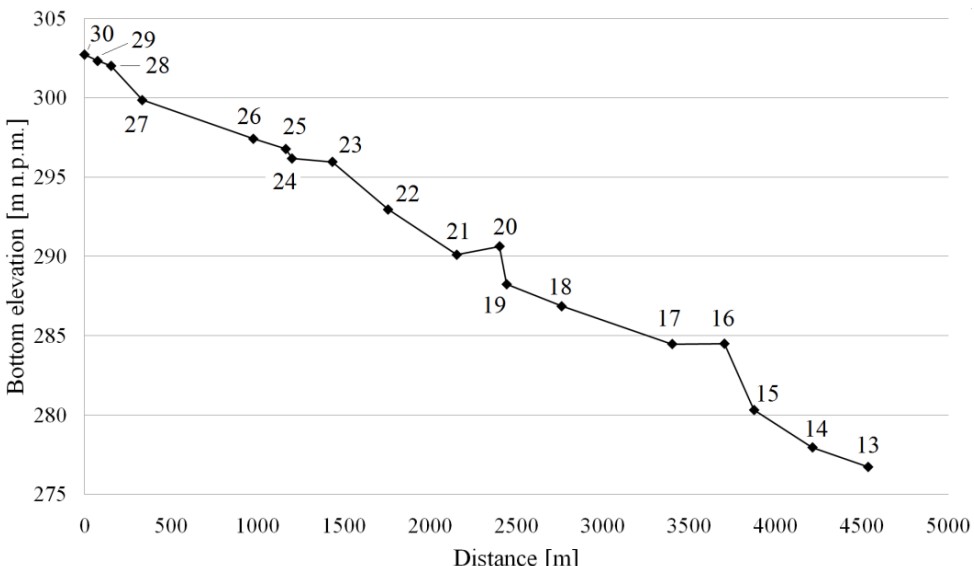

**Figure 3.** Longitudinal profile.

The values of the bottom's gradient fluctuate from negative values to 58‰, with the mean being 5.7‰ for the segment. The measurement takes into consideration local slope changes and the river's morphology, with variable segments represented by a larger density of cross-sections.

### 3.2. Sieve Curves

The selected locations for the measurement of granulation provided information about diversity for the segment characterized by granulation parameters (Table 1). The location and the number of sieve curves depend on local conditions in the riverbed. Five distinctively different segments were recognized and, from each of them, granulation samples were taken.

**Table 1.** Granulation mean diameter and standard deviation.

| Cross-Section | $d_m$ [m] | St. Dev. |
|---|---|---|
| 30–27 | 0.104 | 1.76 |
| 26–22 | 0.076 | 2.88 |
| 21–18 | 0.046 | 2.71 |
| 17–15 | 0.068 | 1.73 |
| 14–13 | 0.043 | 2.05 |

The maximal diameter of the grains $d_{max}$ fluctuates between 0.08 m and 0.19 m, but compared with the respective mean diameters $d_m$, there is no positive relation. Along the longitudinal profile, $d_{max}$ is decreasing with one exception in cross-section 18, possibly due to its best discharge capacity.

### 3.3. Channel Forming Discharge

When assuming predominant flood forces rebuild the existing riverbed structure, the structure remains like that until the moment the next event exceeds the level of channel forming discharge (CFD). The size of this discharge, defined as the event that changes the bottom's structure, can be assumed and analysed by means of changes in the curve of the bottom's granulation [26].

The advantage of gathering four different variables on one chart (Figures 4 and 5) is the possibility of the simultaneous evaluation of the level of real stresses, the state in which they exceed the limit values, changes in granulation, and changes in standard deviation values for the whole sieve curve. When the real stresses are continually increasing, the critical stresses change in the function of the granulometric content and there is growth in the nominal grain size. Exceeding the level of critical stresses, achieving the local maximum of the diameter and the minimum of the standard deviation is the upper limit for the creation of the armouring layer and the further mass movement of debris.

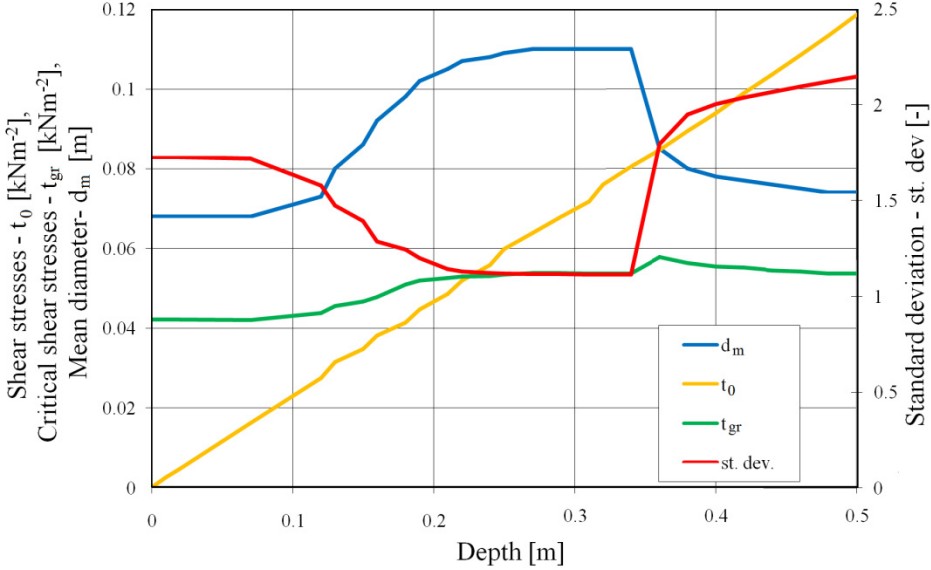

**Figure 4.** Process of creation and erosive destruction of armouring layer, cross-section 16.

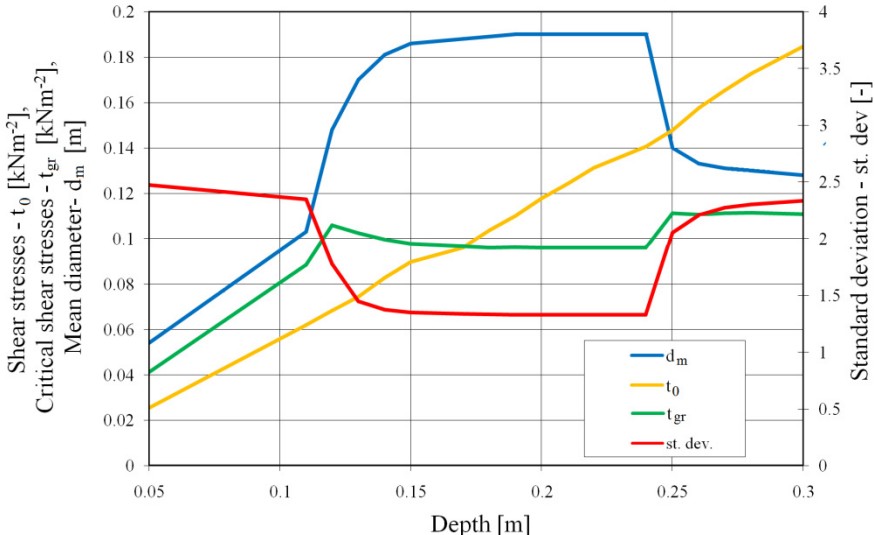

**Figure 5.** Process of creation and erosive destruction of armouring layer, cross-section 20.

In cross-sections 16 and 20, even when the flow is very small, the movement of dragged debris can be expected. Values of CFD in the remaining segments is higher than the bankfull discharge [21], but significantly lower than the flow that reaches the valley.

With the depth of 0.34 m, cross-section 16 achieves the maximal value of diameter for the estimated sieve curve; above these values, a rapid bedload motion in the armouring layer occurs. The hydraulic parameters that represent the conditions of the erosive destruction of the armouring layer are called CFD [11,23]. By this definition, the channel-forming discharge is the value that determines the morphology that is close to actually existing.

The calculated discharges of the segment, obtained by means of the method described above, are included in Table 2. The values of $Q_{max}$ represent the maximal flow capacity of the cross-section, up to the level when the first flood plain is reached; this means stable locations and low slopes. The values of $Q_k$ represent locations with predominant erosion. All the $Q_k$ values are unstable sections, according to the method, because a rise in water level leads to massive erosion faster; the rest ($Q_{max}$) represent stable sections where the valley is flooded but channel-forming discharge is still not reached.

**Table 2.** Channel-forming $Q_k$ and maximal $Q_{max}$ discharges in cross-sections.

| Cross Section | Discharge [m³s⁻¹] | Flow Type | Side of the Valley (R-Right, L-Left) |
|---|---|---|---|
| 30 | 407.95 | $Q_{max}$ | R |
| 29 | 470.02 | $Q_{max}$ | R |
| 28 | 469.98 | $Q_{max}$ | R |
| 27 | 9.64 | $Q_{max}$ | R |
| 26 | 20.61 | $Q_k$ | RLRLR |
| 25 | 469.5 | $Q_{max}$ | RL |
| 24 | 15.8 | $Q_{max}$ | L |
| 23 | 0.59 | $Q_{max}$ | L |
| 22 | 30.26 | $Q_k$ | LR |
| 21 | 146.61 | $Q_{max}$ | RL |
| 20 | 3.5 | $Q_k$ | L |
| 19 | 271.93 | $Q_{max}$ | LR |
| 18 | 42.38 | $Q_k$ | RL |
| 17 | 61.64 | $Q_{max}$ | L |
| 16 | 4.69 | $Q_k$ | LR |
| 15 | 387.78 | $Q_{max}$ | R |
| 14 | 467.81 | $Q_{max}$ | RL |
| 13 | 345.51 | $Q_{max}$ | L |

The $Q_{max}$ values do not offer complete information about the situation in the represented cross-section if the discharge is small. It is still possible for any larger discharge to start the erosion, making this particular cross-section unstable. Erosive discharges were verified in all the stable cross-sections, as higher than a one-hundred-year flood and at least by factor two higher than largest $Q_k = 42.38$ m³s⁻¹ observed in cross-section 18. Why this particular cross-section became erosive can be explained by analysing selected hydraulic parameters, calculated by the model. The setting of parameters linked to each other reveals a significant diversity in the slope, in shear stresses and in flow area at the length of the segment (Figure 6).

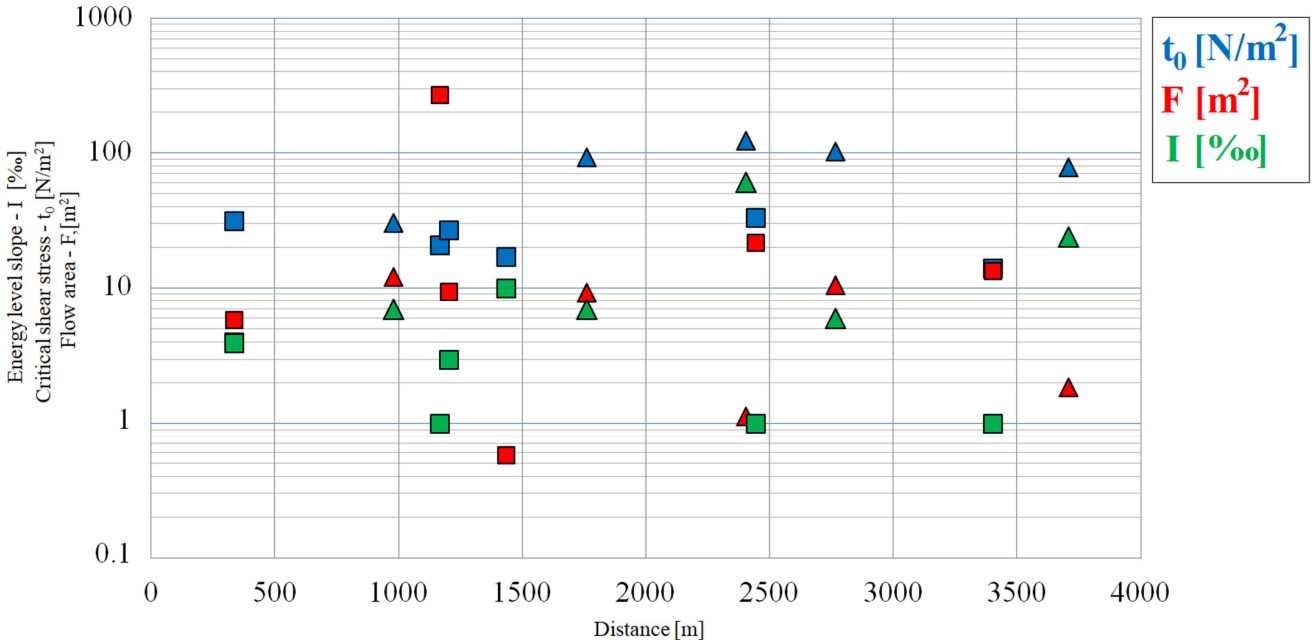

**Figure 6.** Listing of energy level slope, hydraulic radius, velocity (triangle-erosive, square-stable).

The range in the slopes of the energy line is between 1‰ and 61‰. The energy line slope in stable sections is, on average, 3.33‰, while in erosive sections, the slope is 21‰. Shear stresses are also generally higher in erosive cross-sections. The level of shear stress is 51.75 Nm⁻² on average, which, with the size of the bottom granulation $d_m = 0.46$–0.104 m, creates a series of segments alternately susceptible and resistant to the erosion. The phenomenon is not unusual [27]; however, Prądnik is characterized by frequent variability in the length. At the distance of 2800 m (of the measured 4500 m), the CFD will appear frequently in five locations (Table 2), separated by the segments with a predominance of accumulation.

## 4. Discussion

The final results, i.e., the map of stable and erosive sections (Figure 7), is a product of connected data and procedures. The measurement of the river-bed geometry and granulation starts the initial phase of data collection. The numerical model is created, and calibration, based on water surface elevation, and roughness, based on granulation and land cover, is performed. The modelling results form the underlying data for the sieve curve prognosis procedure, which leads to the values of CFD. The rapid fall in granulation (i.e., $d_m$ from 0.11 to 0.074 m in cross-section 16, Figure 4) means that there is, in fact, a breakup of the armouring layer, mass bedload movement, and large erosive changes. This discharge is, then, compared with the cross-section flow capacity (bankfull discharge) where the limit is the vertical range of the floodplain. If the CFD is lower than the bankfull discharge, the cross-section is marked as erosive (unstable) and the $Q_k$ value is presented (5 cross-sections found). Other cross-sections are marked as stable and the values of the

bankfull discharge $Q_{max}$ are collected. Hydraulic parameters can also be presented in relation to the stable and unstable segments if the river training is considered. In that case, decision making will also be supported by the results and there are arguments that suggest why some locations (stable locations) should be excluded.

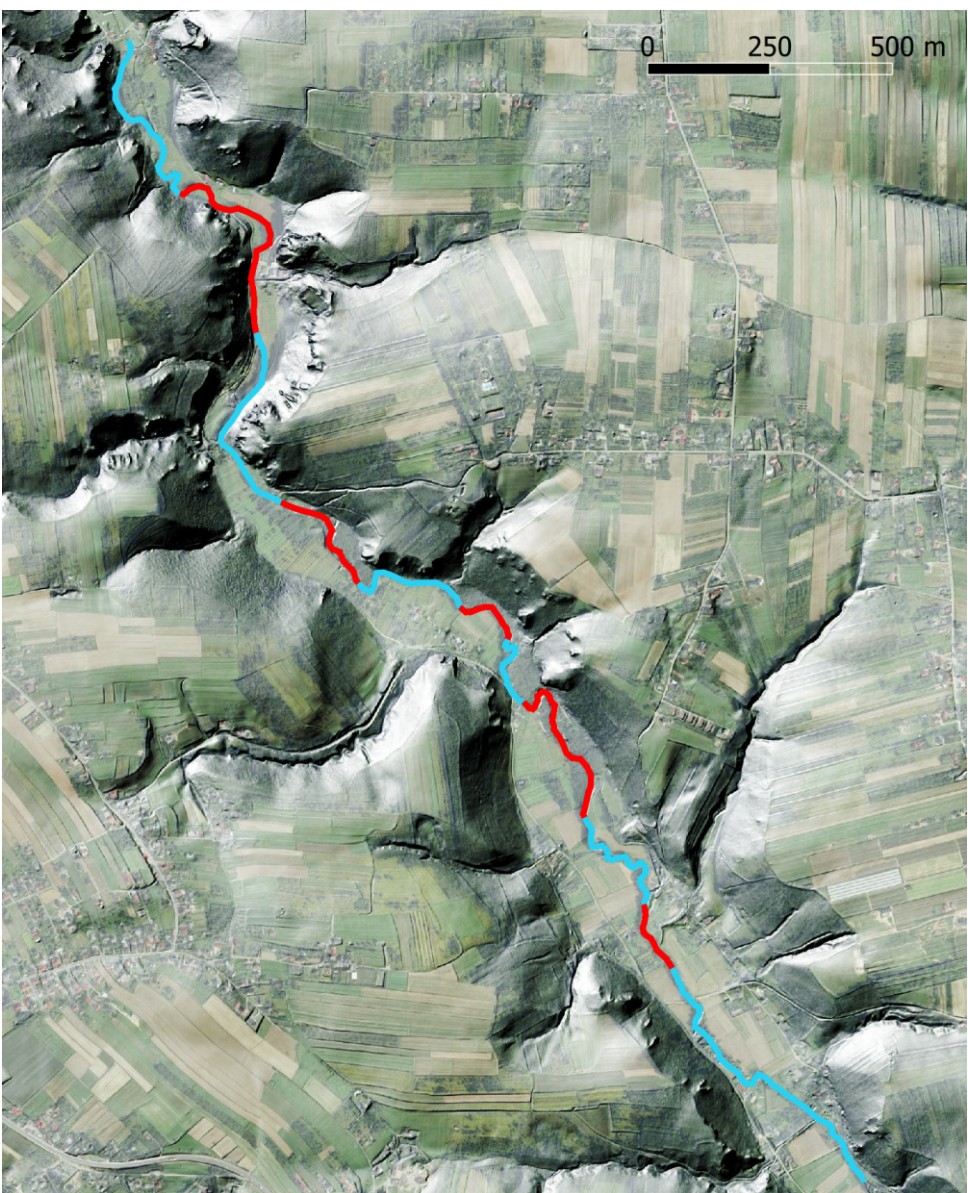

**Figure 7.** Sections: erosive (red) and stable (blue) (source of data: ShadedRelief, ORTO from http://mapy.geoportal.gov.pl WMTS accessed on 13 October 2022).

The cross-section's geometry (Figure 2) adds to the flow capacity and the CFD variability. Prądnik switches valley sides 11 times between cross-sections 14 and 30; this is 4 times on stable segments and 7 times on erosive segments (Figure 7). The river length per meander along the stable segments is 0.187 km and along erosive is 0.346 km. The changes in Prądnik are strong and frequent: the slope is by a factor of 2 or more in at least 11 locations, and the mean diameter is 2.42 and the maximal grain size is 2.38. Multiple attempts were made to establish the optimal locations and the number of cross-sections, until the final number of CFD was established.

Under the effects of simulated floods, the mean diameter $d_m$ rises (i.e., cross-section 16, from 0.068 to 0.11 m), the standard deviation decreases, and the critical shear stresses

increase, but only until the CFD is reached. After the conditions of massive erosion occur, all of these parameters move fast into opposite direction. Since erosive cross-sections are exposed to the process more often than stable ones (CFD at $Q_k = 3.5$–$42.38$ m$^3$s$^{-1}$ and in stable $Q_k > 89$ m$^3$s$^{-1}$) at any given moment, the sieve curve analysis should provide diversified stability conditions, represented by the current effects of the bedload movement (Figures 4 and 5) [8,22,25].

The cross-sections' geometry on the segment shows access to the valley for the floods ranging from 0.059 to 470 m$^3$s$^{-1}$. In general, CFDs are higher than the cross-section's bankfull capacity; under these conditions, a major part of the segment is formed but five cross-sections are erosive. In two cases, the amounts are very small: 3.5 m$^3$s$^{-1}$ and 4.69 m$^3$s$^{-1}$ in cross-sections 16 and 20, respectively. Cross-section 16 is placed directly in the area of the cone-shaped fan on the local side karstic valley. Moreover, the detailed vision in the field reveals that the banks at the segment of 80 m, situated directly above cross-section 26, are double-side strengthened as a result of the protection of the properties and the road parallel to the stream. The total length of the segment susceptible to erosion amounts to 1636 m, which is 35% of the tested distance; indeed, 380 m should be considered as strongly erosive. If river training close to nature is planned, only this segment needs to be considered.

## 5. Conclusions

Channel-forming discharges were calculated using a combined procedure of 1D modelling and a sieve-curve prediction analysis. If a sufficiently large discharge is modelled, erosion will appear in all cross-sections; however, erosive cross-sections display lower capacities by at least a factor of two. Erosive cross-sections also have a larger flow capacity when the floodplain is reached. Both of these factors prove a distinct and measurable difference between the erosive and stable section, which could be used to distinguish them.

A key aspect of the proposed procedure is based on hydraulic modelling, together with establishing channel-forming discharge, flood plain discharge and determining which one is bigger. An extension takes into consideration the specific grain parameters of southern Polish streams, implemented in the sieve curve prediction mechanism. The method can be repeated with only minor adjustments for the regional grain shapes. The density of the cross-section pattern should be corrected until the erosive and stable sections of the segment are finally established.

Summing up, it should be stated that, thanks to using the calculation tools of numeric modelling in hydraulic flow conditions, it is possible to separate the individual cross-sections at risk of erosion with significant accuracy; this can contribute to limiting the range of segments intended for regulation. At the same time, it can be noticed that, alongside the progress made in the accessibility of measurement tools, an increase in calculation capabilities, and the growth in measurement data, the specification of results indicates the increased number of erosive cross-sections in regard to the length of the stream. It follows that created (publicly used) models still do not have sufficient accuracy and the required number of measured cross-sections; this is in accordance with the standards in force that, however, remain an insufficient assessment of the riverbed's erodibility. The prediction of bedload transport, supported by hydraulic modelling, offers a tool to quantify the scope of work and limit the range of interference in the river (in this investigation it is 1/3 of the whole segment). Implementing the sieve curve prediction into the procedure, to support 1D modelling, can answer the question of the river bed stability. What we would like to see is the combined procedure being fully automatic, and attached to the flood risk prediction projects. The innovative approach would be used in three steps: create flood prone zones, verify erosive locations and finally limit river training works. The proposed approach should be supplementary to the design of flood prone/risk zones as a tool to achieve river training, close to the nature standard, by minimizing intervention.

**Author Contributions:** Investigation, J.F. and M.W. All authors have read and agreed to the published version of the manuscript.

**Funding:** Financing of the Ministry of Science and Higher Education as part of statutory activities.

**Institutional Review Board Statement:** Not applicable.

**Informed Consent Statement:** Not applicable.

**Data Availability Statement:** Not applicable.

**Conflicts of Interest:** The authors declare no conflict of interest.

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
