# Peer review of "Procedure of Numerical Modelling and Estimation of Sieve Curve Changes as a Tool to Define Riverbed’s Erodibility"

_sustainability, doi:10.3390/su15021468_

Round 1

Reviewer 1 Report

Dear Authors,

After reading the manuscript “Procedure of numerical modeling and estimation of sieve curve changes as a tool to define riverbed’s erodibility ", I think that the presented research is important, but needs, major revision.

-        Please follow the journal author instructions. It would be useful for the reader to follow the classical text structure (i.e. Introduction-methodology-results-discussion-conclusions). A better presentation of your results and an extensive discussion would improve your paper

  I tried to give some detailed comments, but the line numbers are missing. I appreciate to take these comments into account and please revise the following comments.

Specific comments:

-        Abstract: Please, give a definition of “1-D HEC-RAS” when it mentions for the first time.  

-        Introduction: From my point of view, the article does not provide a sufficiently thorough review of the issue under study. I suggest that the authors should do a better analysis of the literature.

-        I recommend adding a detailed section about “results and discussion”

-        In the conclusion section, the paper lacks a very clear and good justification for what is new and innovative about this case.

Author Response

Dear Mr./Ms. Reviewer

Thank you for giving us the opportunity to submit a revised draft of our manuscript titled “Procedure of numerical modeling and estimation of sieve curve changes as a tool to define riverbed’s erodibility” to Editorial Department of Sustainability Special Issue "River Flood Indicators for Sustainability: Field Studies, Trends and Modeling". We appreciate the time and effort that the reviewers have dedicated to providing valuable feedback on our manuscript. We are grateful to the reviewers for their insightful comments. We have been able to incorporate changes to reflect all of the suggestions provided by You.

Sincerely,
Authors

Reviewer 2 Report

This manuscript presented a study to use the HEC-RAS model to numerically simulate sieve curve changes for the PrÄ…dnik stream which then serves as a tool to define riverbed’s erodibility. The topic of this study fits well in the scope of Sustainability, but ALL sections of the manuscript will need significant improvements before it can be published. 

Firstly, I suggest that the authors revise the Abstract. The purpose of abstracts for research articles is to provide the essence of papers to readers. The major portion of the Abstract of this manuscript discusses the challenges/motivations of this study, however, I don’t see a good summary of any key results.

Secondly, there is no discussion about the uncertainties of the measurements including those for granulation, flow values from gauge stations, and so on. From my perspective, the validity of any observational dataset is only significant with their uncertainty provided.

Thirdly, regarding the one-dimensional hydraulic model HEC-RAS, the authors failed to include in the manuscript both the initial conditions for the model and how the model simulation is validated. 

Last but not least, I could not see a good connection between all figures. The authors discussed each figure individually and then jumped to others without discussing the relationships between them. In addition, Figure 8 is not cited or discussed at all in the manuscript.

Author Response

(The authors gave the same response as above.)

Round 2

Reviewer 1 Report

Well done.